# Fractional Line Integral

**Gabriel Bengochea** [1] and **Manuel Ortigueira** [2,*]

1   Academia de Matemática, Universidad Autónoma de la Ciudad de México, Ciudad de México 09790, Mexico; gabriel.bengochea@uacm.edu.mx
2   CTS-UNINOVA and DEE, NOVA School of Science and Technology, NOVA University of Lisbon, Quinta da Torre, 2829-516 Caparica, Portugal
*   Correspondence: mdo@fct.unl.pt

**Abstract:** This paper proposed a definition of the fractional line integral, generalising the concept of the fractional definite integral. The proposal replicated the properties of the classic definite integral, namely the fundamental theorem of integral calculus. It was based on the concept of the fractional anti-derivative used to generalise the Barrow formula. To define the fractional line integral, the Grünwald–Letnikov and Liouville directional derivatives were introduced and their properties described. The integral was defined for a piecewise linear path first and, from it, for any regular curve.

**Keywords:** fractional integral; Grünwald–Letnikov fractional derivative; fractional line integral; Liouville fractional derivative

**MSC:** Primary 26A33; Secondary 26A42

## 1. Introduction

It is no use to refer to the great evolution that made fractional calculus invade many scientific and technical areas [1–4]. Advances in various aspects of fractional calculus led to a question: Why are there no fractional counterparts of some classic results? In fact, and notwithstanding the progress, there are several singular situations. One of them was, until recently, the non-existence of the definition of the "fractional definite integral". This gap was filled by Ortigueira and Machado [5]. Here, we tried to fill in another gap, by introducing a definition of the fractional line integral. This generalization was motivated by the results presented in [6], where classic theorems of vectorial calculus were introduced, but for integrations over rectangular lines. With the integral introduced here, the Green theorem, for example, can be generalised. Here, we took advantage of the results stated in [5] to propose a fractional line integral. We used directional derivatives. However, and according to the considerations made in [5], not all the directional derivatives are suitable for our purposes. We opted for the directional derivatives resulting from the generalization of the Grünwald–Letnikov (GL) and Liouville (L) directional derivatives. These were introduced and their main properties listed. It is important to refer to the presentation of the Liouville directional regularised derivative.

The classic definite Riemann integral served as guide to define the fractional definite integral, since we required them to share the same properties. The fractional definite integral was expressed in terms of the anti-derivative generalising the Barrow formula.

To introduce the fractional line integral (FLI), we started by defining it on a segment of an oriented straight-line. This procedure was enlarged to a piecewise linear path formed by a sequence of connected straight-line segments. By using a standard procedure consisting of approximating a curve by a sequence of piecewise linear paths, we introduced the integration over any simple rectifiable line. Its main properties are presented.

The paper is outlined as follows. In Section 2, we introduce the required background. In Section 3, we describe the GL and L directional derivatives that are the basis for defining

the fractional line integral. The corresponding properties are also presented. In Section 4, we introduce the fractional line integral and its main properties. Finally, we draw some conclusions.

## 2. Background

### 2.1. Functional Framework

The theory we develop below is expected to be useful in generalising the classic vectorial theorems suitable for dealing with fractionalisations of important equations of physics, as is the case of the Maxwell equations [6,7]. Therefore, we needed a framework involving functions $f(t)$, $t \in \mathbb{R}$, which are of exponential order, to have the bilateral Laplace transform (BLT), or absolutely or square integrable, to have the Fourier transform (FT). To ensure that the left derivatives we used exist, we assumed also that they:

1. are almost everywhere continuous;
2. have bounded variation;
3. verify:

$$|f(x)| < A \frac{1}{|x|^{\gamma+1}}, \ \gamma, A \in \mathbb{R}^+, \text{ for } \ x < x_0 \in \mathbb{R}. \tag{1}$$

In particular, we can have $f(x) \equiv 0$, $x < x_0$ (this is called a right-hand function).

**Remark 1.** *If a given function has bounded support, we extend it to $\mathbb{R}$, with a null extrapolation. This maintains the maximum generality of our framework.*

### 2.2. Suitable Fractional Derivatives

When performing the introduction of the fractional definite integral [5], a discussion about the problem of the definition of the fractional derivative (FD) suitable for such an objective was performed. It was shown that among the many definitions, we should require one-sided derivatives defined on $\mathbb{R}$ in order to keep several classic relations valid. The Grünwald–Letnikov (GL) [4,8,9] and (regularised) Liouville (rL) derivatives [4,10] were adopted. The forward Grünwald–Letnikov and Liouville derivatives are given respectively by:

$$D^{\alpha} f(x) = \lim_{h \to 0^+} \frac{\sum\limits_{n=0}^{\infty} (-1)^n \frac{(-\alpha)_n}{n!} f(x - nh)}{h^{\alpha}}, \tag{2}$$

where $(a)_n = \prod_{k=0}^{n-1}(a+k)$, $((a)_0 = 1)$ represents the Pochhammer symbol for the raising factorial, and:

$$D_L^{\alpha} f(x) = \frac{1}{\Gamma(-\alpha)} \int_0^{\infty} \left[ f(x-\tau) - u(\alpha) \sum_{n=0}^{N} \frac{(-1)^n f^{(n)}(x)}{n!} \tau^n \right] \tau^{-\alpha-1} d\tau, \tag{3}$$

where $N$ is the integer part of $\alpha$, so that $\alpha - 1 < N \le \alpha$, $N \in \mathbb{Z}_0^+$, and $u(\alpha)$ is the Heaviside unit step to ensure that the summation only exists for positive $\alpha$. The conditions stated in the previous subsection, namely (1), are sufficient for the existence of the above derivatives. Despite our work being based in the derivatives GL and rL, we included one version of the directional derivative for the usual Liouville derivative (L) [9,11]:

$$^{RL}D_+^{\alpha} f(x) := \frac{1}{\Gamma(m-\alpha)} \frac{d^m}{dx^m} \int_{-\infty}^{x} (x-\xi)^{m-\alpha-1} f(\xi) d\xi, \tag{4}$$

and the Liouville–Caputo derivative (LC) [1]:

$$^{LC}D_+^{\alpha} f(x) := \frac{1}{\Gamma(m-\alpha)} \int_{-\infty}^{x} (x-\xi)^{m-\alpha-1} \frac{d^m}{d\xi^m} f(\xi) d\xi, \tag{5}$$

where $m - 1 < \alpha \le m$, $m \in \mathbb{Z}^+$. These two derivatives could be alternatives to GL or rL.

**Remark 2.** *The concept of "forward" is tied here to the causality in the sense of "going from past to future." This implies not only an order, but also a direction on the real line.*

The study of the equivalence of the two derivatives was performed in [10]. It is not difficult to show that they are really equivalent for functions with the BLT or FT. We only have to compute the derivatives of the exponential $e^{st}$, $s \in \mathbb{C}, t \in \mathbb{R}$ [8]. This makes it easier to justify some of the relevant characteristics they enjoy, namely the index law [8,10]. This property means that for a given FD of order $\alpha > 0$, there is an FD, of negative order, that we call the "anti-derivative" that is left and right inverse of the FD:

$$D^\alpha D^{-\alpha} f(x) = D^{-\alpha} D^\alpha f(x) = f(x). \tag{6}$$

It can be shown [4] that the GL, rL, LC, and FD of the non-null constant function are identically null. The L derivative of such a function does not exist, since the integral is divergent. It must be stressed here that, for negative order (anti-derivative), the rL, L, and LC are equal:

$$D_L^\alpha f(x) = \frac{1}{\Gamma(-\alpha)} \int_0^\infty f(x-\tau)\tau^{-\alpha-1}d\tau = -\frac{1}{\Gamma(-\alpha+1)} \int_0^\infty f(x-\tau)d\tau^{-\alpha}, \quad \alpha < 0. \tag{7}$$

**Remark 3.** *Everything what was performed here can be replicated for the backward derivatives. We have not done so, since this is not very interesting.*

*2.3. Order-One Definite Integral*

Consider a closed interval $[a,b] \subset \mathbb{R}$ where $f(x)$ is continuous. There are several ways of introducing the definite integral [12,13]. Probably the simplest is through the Riemann sum.

**Definition 1.** *Divide the interval $[a,b]$ into N small intervals with lengths $\Delta_i$, $i = 1, 2, \ldots, N$. Let $\xi_i \in \Delta_i$ ($i = 1, 2, \ldots, N$). We call the definite integral of $f(x)$ over $[a,b]$ the limit:*

$$S = \lim_{\max(\Delta_i) \to 0} \sum_{i=1}^N f(\xi_i)\Delta_i \tag{8}$$

*The sum S is represented by $\int_a^b f(\xi)d\xi$ [12].*

Using the time scale approach, we can define a nabla derivative and its inverse [4]. This is given by:

$$f^{(-1)}(x) = \lim_{\sup(\Delta_i) \to 0} \sum_{i=0}^\infty \Delta_i f(x - h_i)$$

where $h_i = \sum_{k=0}^{i-1} \Delta_k$ ($h_0 = 0$). With this anti-derivative, we can rewrite (8) as:

$$\int_a^b f(x)dx = f^{(-1)}(b) - f^{(-1)}(a). \tag{9}$$

For simplicity and because the function is continuous, we adopted a procedure, frequently used in practice, that consisted of using equal length intervals, $\Delta_i = \frac{b-a}{N} = h$, $i = 1, 2, \ldots, N$, and $\xi_i$ the first or last point in each one, so that we can set, for example, $\xi_i = a + (i-1)h = b - ih$, $i = 1, 2, \ldots, N$. We have, then:

$$\int_a^b f(x)dx = \lim_{h \to 0} h \sum_{i=0}^{N-1} f(a+ih) = \lim_{h \to 0} h \sum_{i=1}^N f(b-ih). \tag{10}$$

*2.4. Fractional Definite Integral*

A generalisation of the concept of the definite integral must conform with its properties. The first approach to obtain a fractional definite integral was performed in [5]. Here, we present a slightly different definition.

**Definition 2.** *We defined the α-order fractional integral (FI) of $f(x)$ ($f(-\infty) = 0$) over the interval $(-\infty, a)$ by:*

$$I^\alpha f(-\infty, a) = \frac{1}{\Gamma(\alpha)} \int_0^\infty f(a - \tau)\tau^{\alpha-1}d\tau = \frac{1}{\Gamma(\alpha + 1)} \int_0^\infty f(a - \tau)d\tau^\alpha = f^{(-\alpha)}(a). \quad (11)$$

For simplification, we used the notation:

$$I^\alpha f(-\infty, a) = \int_{-\infty}^a f(x)dx^\alpha.$$

**Theorem 1.** *If $b > a$, then:*

$$I^\alpha f(a, b) = \int_a^b f(x)dx^\alpha = f^{(-\alpha)}(b) - f^{(-\alpha)}(a). \quad (12)$$

In fact, this relation must be valid, in order to keep the formula valid:

$$\int_a^b f(x)dx = \int_a^c f(x)dx + \int_c^b f(x)dx,,$$

with $a < c < b$. The relation (12) is none other than the *fractional Barrow formula*. The definition of the fractional definite integral (12) is consistent with the *fractional fundamental theorem of integral calculus* (integer order):

**Theorem 2.**

$$I^\alpha D^\alpha g(a, x) = D^{-\alpha} D^\alpha g(x) - D^{-\alpha} D^\alpha g(a) = g(x) - g(a), \quad (13)$$

*and*

$$D^\alpha[I^\alpha g(a, x)] = g(x). \quad (14)$$

These results come immediately from the properties of the rL (or GL) derivative. In particular, the derivative of a constant is zero.

## 3. The Grünwald–Letnikov and Liouville Directional Derivatives

The usefulness, advantages, and properties of the GL and rL derivatives, introduced above, were studied in [5,10]. Here, we present their directional formulations.

**Definition 3.** *Consider a function $f(\mathbf{x})$, $\mathbf{x} \in \mathbb{R}^n$, and let $\mathbf{v} \in \mathbb{R}^n$ be a unitary vector defining the direction of the derivative computation and the half-straight line:*

$$\{\boldsymbol{\xi} : \boldsymbol{\xi} = (\mathbf{x} - kh\mathbf{v}), \ h \in \mathbb{R}^+, \mathbf{x} \in \mathbb{R}^n \ k \in \mathbb{N}_0\}. \quad (15)$$

*Consider a continuous function, $f(\mathbf{x})$, such that $|f(\mathbf{x} - k\mathbf{v})|$ decreases at least as $\frac{1}{k^{|\alpha|+1}}$, when $k \to \infty$ [4]. We defined the GL directional derivative as:*

$$D_{\mathbf{v}}^\alpha f(\mathbf{x}) = \lim_{h \to 0} h^{-\alpha} \sum_{k=0}^\infty \frac{(-\alpha)_k}{k!} f(\mathbf{x} - kh\mathbf{v}), \quad (16)$$

*where again, $(a)_k$ represents the Pochhammer symbol for the rising factorial.*

The relation of the GL and Liouville derivatives studied in [10] led us to introduce an analogous definition for the directional case, considering the general regularised case, not presented elsewhere.

**Definition 4.** *We defined the Liouville directional integral (anti-derivative) by:*

$$D_{\mathbf{v}}^{\alpha} f(\mathbf{x}) = \frac{1}{\Gamma(-\alpha)} \int_0^{\infty} v^{-\alpha-1} f(\mathbf{x} - v\mathbf{v}) \, dv, \tag{17}$$

*where $\alpha < 0$.*

As is well known, when $\alpha > 0$, the above integral is singular [9]. However, it can be regularised through the procedure followed in [10] to obtain:

$$D_{\mathbf{v}}^{\alpha} f(\mathbf{x}) = \frac{1}{\Gamma(-\alpha)} \int_0^{\infty} \left( f(\mathbf{x} - v\mathbf{v}) - \sum_{m=0}^{N} \frac{1}{m!} \frac{d^m f(\mathbf{x} - v\mathbf{v})}{dv^m} \Big|_{v=0} v^m \right) v^{-\alpha-1} \, dv, \tag{18}$$

where $N = \lfloor \alpha \rfloor$ is the greatest integer less than or equal to $\alpha$.

**Definition 5.** *We defined the Liouville directional derivative (L) by:*

$$^{RL}D_{\mathbf{v}}^{\alpha} f(\mathbf{x}) = \frac{1}{\Gamma(m-\alpha)} D_{\mathbf{v}}^m \int_0^{\infty} v^{m-\alpha-1} f(\mathbf{x} - v\mathbf{v}) \, dv, \tag{19}$$

*where $D_{\mathbf{v}}^m$ means to apply m-times the usual directional derivative in the direction $\mathbf{v}$, $m - 1 < \alpha \leq m$ and $m \in \mathbb{Z}^+$.*

**Definition 6.** *We defined the Liouville–Caputo derivative (LC) by:*

$$^{LC}D_{\mathbf{v}}^{\alpha} f(\mathbf{x}) = \frac{(-1)^m}{\Gamma(m-\alpha)} \int_0^{\infty} v^{m-\alpha-1} \frac{d^m}{dv^m} f(\mathbf{x} - v\mathbf{v}) \, dv, \tag{20}$$

*where $m - 1 < \alpha \leq m$ and $m \in \mathbb{Z}^+$.*

To test the coherence of the definitions, take the exponential $f(\mathbf{x}) = e^{\mathbf{s} \cdot \mathbf{x}}$, $\mathbf{x} \in \mathbb{R}^n$, where $(\mathbf{s} \cdot \mathbf{x})$ is the inner product of the constant $\mathbf{s} \in \mathbb{C}^n$ and $\mathbf{x}$. Inserting $f(\mathbf{x})$ into relations (16) to (20), we got [9]

$$D_{\mathbf{v}}^{\alpha} e^{\mathbf{s} \cdot \mathbf{x}} = (\mathbf{s} \cdot \mathbf{v})^{\alpha} e^{\mathbf{s} \cdot \mathbf{x}}, \qquad Re(\mathbf{s}) > \mathbf{0}. \tag{21}$$

To obtain this result, we had to use the general integral representation for the Gamma function defined on $\mathbb{C} \setminus \mathbb{Z}_0^-$ [14]:

$$\Gamma(\alpha) = \int_0^{\infty} \left[ e^{-\tau} - u(-\alpha) \sum_{n=0}^{N} \frac{(-1)^n}{n!} \tau^n \right] \tau^{\alpha-1} \, d\tau,$$

where $N$ is the greatest integer less than $-\alpha$ if $\alpha < 0$ and $u(\cdot)$ is the Heaviside function.

The expression $Re(\mathbf{s}) > \mathbf{0}$ means that each component $s_j, j = 1, 2, \ldots, n$ of $\mathbf{s}$ has positive real parts. It is a sufficient condition for the convergence of the series in (16) or of the integrals in (17) to (20). Relation (21) is in agreement with similar results in [9] and means that:

1.  The exponential is the eigenfunction of the introduced derivatives;
2.  The BLT, $F(\mathbf{s})$ of the derivatives of $f(\mathbf{x})$, is given by:

$$\mathcal{L}[D_{\mathbf{v}}^{\alpha} f(\mathbf{x})] = (\mathbf{s} \cdot \mathbf{v})^{\alpha} F(\mathbf{s}), \qquad Re(\mathbf{s}) > \mathbf{0}. \tag{22}$$

where $F(\mathbf{s}) = \mathcal{L}[f(\mathbf{x})]$. With $\mathbf{s} = i\boldsymbol{\omega}$, we obtained the same result for the Fourier transform.

The main properties of the above derivatives are easily deduced from (17) to (20); see, for example [8].

1.  Linearity vectorial functions: Let $\mathbf{f}(\mathbf{x}) = f_1(\mathbf{x})\mathbf{e}_1 + f_2(\mathbf{x})\mathbf{e}_2 + f_3(\mathbf{x})\mathbf{e}_3$. Then:

$$D_{\mathbf{v}}^{\alpha}[f(\mathbf{x}) + g(\mathbf{x})] = D_{\mathbf{v}}^{\alpha}f(\mathbf{x}) + D_{\mathbf{v}}^{\alpha}g(\mathbf{x}). \tag{23}$$

2.  Commutativity and additivity of the orders
    If $\alpha, \beta \in \mathbf{R}$

$$D_{\mathbf{v}}^{\alpha}\left[D_{\mathbf{v}}^{\beta}f(\mathbf{x})\right] = D_{\mathbf{v}}^{\beta}[D_{\mathbf{v}}^{\alpha}f(\mathbf{x})] = D_{\mathbf{v}}^{\alpha+\beta}f(\mathbf{x}). \tag{24}$$

3.  Neutral and inverse elements:
    In the previous property, we can set $\alpha + \beta = 0$; therefore, the inverse derivative exists and can be obtained by using the same formula.

$$D_{\mathbf{v}}^{\alpha}\left[D_{\mathbf{v}}^{-\alpha}f(\mathbf{x})\right] = D_{\mathbf{v}}^{0}f(\mathbf{x}) = f(\mathbf{x}). \tag{25}$$

4.  Rotation:
    Suppose that an invertible matrix $\mathbf{A}$ exists such that we can perform the variable change $\mathbf{A}\mathbf{x}$ for $\mathbf{x}$ and $\mathbf{v}$ is not the zero vector. Then:

$$\begin{aligned}
D_{\mathbf{v}}^{\alpha}f(A\mathbf{x}) &= \frac{1}{\Gamma(-\alpha)} \int_0^{\infty} u^{-\alpha-1}f(A[\mathbf{x} - u\mathbf{v}]) \, du \\
&= \frac{1}{\Gamma(-\alpha)} \int_0^{\infty} u^{-\alpha-1}f(A\mathbf{x} - uA\mathbf{v}) \, du.
\end{aligned} \tag{26}$$

As the matrix $A$ is invertible and $\mathbf{v}$ is a non-null vector, then $\|A\mathbf{v}\| \neq 0$. As $uA\mathbf{v} = u\|A\mathbf{v}\|\frac{A\mathbf{v}}{\|A\mathbf{v}\|}$, we introduced $\xi = \|A\mathbf{v}\|u$ and $\mathbf{w} = \frac{A\mathbf{v}}{\|A\mathbf{v}\|}$ to obtain:

$$D_{\mathbf{v}}^{\alpha}f(A\mathbf{x}) = \frac{\|A\mathbf{v}\|^{\alpha}}{\Gamma(-\alpha)} \int_0^{\infty} w^{-\alpha-1}f(A\mathbf{x} - \xi\mathbf{w})) d\xi, \tag{27}$$

which leads to:

$$D_{\mathbf{v}}^{\alpha}f(A\mathbf{x}) = \|A\mathbf{v}\|^{\alpha} D_{\mathbf{w}}^{\alpha}f(\mathbf{x}')\big|_{\mathbf{x}'=A\mathbf{x}'} \tag{28}$$

generalising also a classic result.

## 4. On the Fractional Line Integrals

The introduction of the notion of the fractional definite integral was performed in [5] and reformulated above in Section 2.4. Here, we reproduced such a definition, but with a vectorial representation.

Let $f(\mathbf{x})$, $\mathbf{x} \in \mathbb{R}^n$, such that its directional derivatives of any order exist. We assumed the canonical base $\mathbf{e_j}$, $j = 1, 2, \ldots, n$. Let us denote their anti-derivatives along the $x_1$ axis by $f_{\mathbf{e_1}}^{(-\alpha)}(\mathbf{x})$.

**Definition 7.** *Let $\alpha > 0$. We defined the $\alpha$-order fractional integral of $f(\mathbf{x})$ over the interval $(-\infty, a)$ on the $x\mathbf{e_1}$ axis through:*

$$I_{\mathbf{e_1}}^{\alpha} f_{\mathbf{x}}(-\infty, \mathbf{a}) = \int_{-\infty}^{a} f(x\mathbf{e_1}) dx^{\alpha} = f_{\mathbf{e_1}}^{(-\alpha)}(\mathbf{a}), \tag{29}$$

*and:*

$$I_{\mathbf{e_1}}^{\alpha} f_{\mathbf{x}}(\mathbf{a}, \mathbf{b}) = \int_{a}^{b} f(x\mathbf{e_1}) dx^{\alpha} = f_{\mathbf{e_1}}^{(-\alpha)}(\mathbf{b}) - f_{\mathbf{e_1}}^{(-\alpha)}(\mathbf{a}), \tag{30}$$

*where $(\mathbf{a}, \mathbf{b}) = (a\mathbf{e_1}, b\mathbf{e_1})$.*

Using the expression of the Liouville anti-derivative, we can write:

$$I^\alpha_{\mathbf{e_1}} f_{\mathbf{x}}(\mathbf{a}, \mathbf{b}) = \frac{1}{\Gamma(\alpha)} \int_0^\infty [f(b\mathbf{e_1} - x\mathbf{e_1}) - f(a\mathbf{e_1} - x\mathbf{e_1})] dx^\alpha. \tag{31}$$

From the standard (integer order) Barrow formula $\int_a^b f'(x)dx = f(b) - f(a)$, we derived the expression:

$$I^\alpha_{\mathbf{e_1}} f_{\mathbf{x}}(\mathbf{a}, \mathbf{b}) = \frac{1}{\Gamma(\alpha)} \int_0^\infty \int_a^b D^1_{\mathbf{e_1}} f(y\mathbf{e_1} - x\mathbf{e_1}) dy dx^\alpha. \tag{32}$$

Using (24), we obtained:

$$I^\alpha_{\mathbf{e_1}} f_{\mathbf{x}}(\mathbf{a}, \mathbf{b}) = \int_a^b f^{(-\alpha+1)}_{\mathbf{e_1}}(x\mathbf{e_1}) dx. \tag{33}$$

If the integration path, instead of one base axis, is any straight line, defined by a vector $\mathbf{v}$, we need to generalize the above procedure. Consider a scalar field $f : \mathbb{D} \subset \mathbb{R}^n \to \mathbb{R}$ and two points $\mathbf{a}, \mathbf{b} \in \mathbb{D}$, defining the vector $\mathbf{v} = \frac{\mathbf{b} - \mathbf{a}}{\|\mathbf{b} - \mathbf{a}\|}$. The most natural way to define the fractional line integral over a straight-line segment from $\mathbf{a}$ to $\mathbf{b}$ is:

$$I^\alpha_{\mathbf{v}} f_{\mathbf{x}}(\mathbf{a}, \mathbf{b}) = \int_a^b f(x\mathbf{v}) dx^\alpha = f^{(-\alpha)}_{\mathbf{v}}(\mathbf{b}) - f^{(-\alpha)}_{\mathbf{v}}(\mathbf{a}). \tag{34}$$

This definition can be generalized to the case when the integration path $\gamma$ is a sequence of $N$ connected straight-line segments $\gamma_k$, $k = 0, 1, 2, \ldots, N - 1$ with initial and final points $\mathbf{a}_k, \mathbf{a}_{k+1} = \mathbf{b}_k$, respectively ($\mathbf{a}_0 = \mathbf{a}$, $\mathbf{a}_N = \mathbf{b}$), by:

$$I^\alpha_\gamma f = \sum_{k=0}^{N-1} \int_{\mathbf{a}_k}^{\mathbf{a}_{k+1}} f(x\mathbf{v}_k) dx^\alpha = \sum_{k=0}^{N-1} \left[ f^{(-\alpha)}_{\mathbf{v}_k}(\mathbf{a}_{k+1}) - f^{(-\alpha)}_{\mathbf{v}_k}(\mathbf{a}_k) \right], \tag{35}$$

where $\mathbf{v}_k = \frac{\mathbf{a}_{k+1} - \mathbf{a}_k}{\|\mathbf{a}_{k+1} - \mathbf{a}_k\|}$. In Figure 1, we illustrate the way the directional derivative computations were performed. The previous definition can be more general. For this, consider that $\gamma$ is a rectifiable curve. We constructed a sequence of straight-line segments $\bar{\gamma}$ approximating the curve $\gamma$, such that the initial and final points coincided (see Figure 2). For each segment, set $\mathbf{b}_k = \mathbf{a}_{k+1} = \mathbf{a}_k + h_k \mathbf{v}_k$, where $h_k > 0$ is the length of the $k$th segment. If we consider several possible approximants, $\bar{\gamma}_n$, $n = 1, 2, \ldots$ for the curve and let $h_{max}$ be the maximum length in each approximant, then we obtained the expression for the fractional line integral:

$$I^\alpha_\gamma f = \int_\gamma f(x\mathbf{v}) dx^\alpha = \lim_{h_{max} \to 0} \sum_{k=0}^{N-1} \left[ f^{(-\alpha)}_{\mathbf{v}_k}(\mathbf{a}_k + h_k \mathbf{v}_k) - f^{(-\alpha)}_{\mathbf{v}_k}(\mathbf{a}_k) \right]$$

$$= \lim_{h_{max} \to 0} \sum_{k=0}^{N-1} \left[ f^{(-\alpha)}_{\mathbf{v}_k}(\mathbf{b}_k) - f^{(-\alpha)}_{\mathbf{v}_k}(\mathbf{b}_k - h_k \mathbf{v}_k) \right]. \tag{36}$$

If we suppose that $f^{(-\alpha)}(\mathbf{x})$ is differentiable in the curve $\gamma$, then:

$$\left[ f^{(-\alpha)}_{\mathbf{v}_k}(\mathbf{b}_k) - f^{(-\alpha)}_{\mathbf{v}_k}(\mathbf{b}_k - h_k \mathbf{v}_k) \right] = f^{(-\alpha+1)}_{\mathbf{v}_k}(\mathbf{b}_k) h_k + \eta(h_k),$$

with $\lim_{h_{max} \to 0} \eta(h_k) = 0$. From the above, we were led to the following definition.

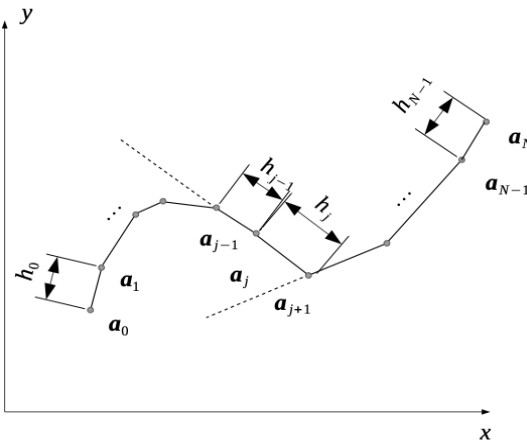

**Figure 1.** Broken line integration path.

**Definition 8.** *Let $\gamma$ be an rectifiable curve by straight-line segments with initial and final points $\mathbf{a}_k, \mathbf{b}_k = \mathbf{a}_{k+1} = \mathbf{a}_k + h_k \mathbf{v}_k$, respectively, where $h_k > 0$ is the length of the kth segment and $\mathbf{v}_k = \frac{\mathbf{a}_{k+1} - \mathbf{a}_k}{\|\mathbf{a}_{k+1} - \mathbf{a}_k\|}$. Suppose that $f^{(-\alpha)}(\mathbf{x})$ is differentiable in a domain $\mathbb{D} \subset \mathbb{R}^n$ with $\gamma \subset \mathbb{D}$. We defined the fractional line integral on the curve $\gamma$ by:*

$$I_\gamma^\alpha f = \lim_{N \to \infty} \sum_{k=0}^{N-1} f_{\mathbf{v}_k}^{(-\alpha+1)}(\mathbf{a}_k) h_k. \tag{37}$$

*When the limit exists, it is denoted by $\int_\gamma f(x\mathbf{v}) dx^\alpha$.*

Suppose that the curve $\gamma$ in Definition 8 can be parametrized by a curve $C^1$, $\mathbf{r} : [a, b] \leftarrow \mathbb{R}^n$. It is clear that the shorter the straight-line segments are, the better the approximation of the curve $\gamma$ is; see Figure 2. In such a situation, in (37), we have that:

- $h_i \approx h_j$;
- $\mathbf{v}_k$ is approximately tangent to $\gamma$.

It follows that the limit in (37) becomes in the usual line integral:

$$\lim_{N \to \infty} \sum_{k=0}^{N-1} f_{\mathbf{v}_k}^{(-\alpha+1)}(\mathbf{a}_k) h_k = \int_\gamma f_{\mathbf{v}}^{(-\alpha+1)}(\mathbf{x}) ds, \tag{38}$$

where $\mathbf{v}$ is a unitary tangent vector at each point $\mathbf{x}$ of the curve $\gamma$. Because the curve $\gamma$ can be parametrized by $\mathbf{r}(u)$ and by the properties of the usual line integral, we obtained that $\mathbf{v} = \mathbf{r}'(u) / \|\mathbf{r}'(u)\|$ and:

$$\int_\gamma f_{\mathbf{v}}^{(-\alpha+1)}(\mathbf{x}) ds = \int_a^b f_{\mathbf{v}}^{(-\alpha+1)}(\mathbf{r}(u)) \|\mathbf{r}'(u)\| du. \tag{39}$$

Suppose that the parametrization $\mathbf{r}(u)$ of $\gamma$ can be written in terms of two distinct parameters $u$ and $\tau$ and that $u_0 \le u \le u_1$ and $\tau_0 \le \tau \le \tau_1$. Hence:

$$\int_{u_0}^{u_1} f_{\mathbf{v}}^{(-\alpha+1)}(\mathbf{r}(u)) \|\mathbf{r}'(u)\| du = \int_{\tau_0}^{\tau_1} f_{\mathbf{v}}^{(-\alpha+1)}(\mathbf{r}(\tau)) \|\mathbf{r}'(\tau)\| d\tau. \tag{40}$$

It follows that the fractional line integral (39) does not depend of the parametric representation of $\gamma$.

**Remark 4.** *In (39), the fractional directional derivative, $f_{\mathbf{v}}^{(-\alpha+1)}(\mathbf{r}(u))$, was taken in order to $\mathbf{r}$, not to $u$.*

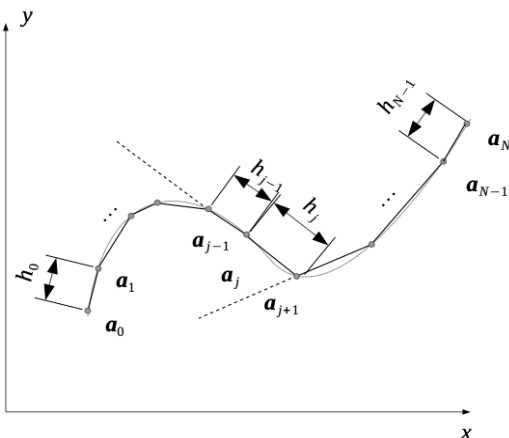

**Figure 2.** Rectifiable integration path.

**Example 1.** *In the case when $\gamma$ is a circumference of radius $R$ centred at the origin,* $\mathbf{r}(u) = (R \cos u, R \sin u), 0 \leq u \leq 2\pi, (39)$ *acquires the form:*

$$I_\gamma^\alpha f = R \int_0^{2\pi} f_{\mathbf{v}}^{(-\alpha+1)}(R \cos u, R \sin u) du, \tag{41}$$

*with $\mathbf{v} = (-\sin u, \cos u)$. By the definition of the directional derivative, we obtained that:*

$$
\begin{aligned}
I_\gamma^\alpha f &= \frac{R}{\Gamma(\alpha-1)} \int_0^{2\pi} \int_0^\infty v^\alpha f(R \cos u + v \sin u, R \sin u - v \cos u) dv du \\
&= \frac{R}{\Gamma(\alpha-1)} \int_0^\infty v^\alpha \int_0^{2\pi} f(R \cos u + v \sin u, R \sin u - v \cos u) du dv
\end{aligned}
\tag{42}
$$

*If we define $\zeta(u) = (R \cos u + v \sin u, R \sin u - v \cos u), 0 \leq u \leq 2\pi$, then:*

$$
\begin{aligned}
||\zeta'(u)|| &= ||(-R \sin u + v \cos u, R \cos u + v \sin u)|| \\
&= \sqrt{(-R \sin u + v \cos u)^2 + (R \cos u + v \sin u)^2} \\
&= \sqrt{R^2 + v^2}
\end{aligned}
\tag{43}
$$

*Therefore,*

$$I_\gamma^\alpha f = \frac{R}{\Gamma(\alpha-1)} \int_0^\infty \frac{v^\alpha}{\sqrt{R^2 + v^2}} \left[ \int_0^{2\pi} f(\zeta(u)) ||\zeta'(u)|| du \right] dv. \tag{44}$$

**Remark 5.** *The integral between brackets in (44) is known as the integral of the line relative to the arc length [13]. If $f$ represents the mass of a thin wire $\zeta(u)$ per unit length, then $\int_0^{2\pi} f(\zeta(u)) ||\zeta'(u)|| du$ is the total mass $M_v$ of the wire. Therefore:*

$$I_\gamma^\alpha f = \frac{R}{\Gamma(\alpha-1)} \int_0^\infty \frac{v^\alpha M_v}{\sqrt{R^2 + v^2}} dv. \tag{45}$$

The line integral (37) has some interesting properties that are easily deduced:

1. Linearity:

$$I_\gamma^\alpha(cf + dg) = cI_\gamma^\alpha f + dI_\gamma^\alpha g, \tag{46}$$

with $c$ and $d$ constants;

2. Additivity:
   Let $\gamma_1$ and $\gamma_2$ be two disjoint lines. If $\gamma = \gamma_1 \bigcup \gamma_2$, then $I_\gamma^\alpha f = I_{\gamma_1}^\alpha f + I_{\gamma_2}^\alpha f$;

3. Orientation:
   Let $\gamma$ be the curve $\mathbf{r}(u), a \leq u \leq b$. The change in the orientation was obtained in the

fractional derivative computation by reversing the tangent vector and the integration limits. Hence:

$$I^\alpha_{-\gamma}f = \int_b^a f^{(-\alpha+1)}_{-\mathbf{v}}(\mathbf{r}(u))\|\mathbf{r}'(u)\|du. \tag{47}$$

While in the $\alpha = 1$ case, $I^\alpha_{-\gamma}f = -I^\alpha_\gamma f$, this may not happen in the fractional case, since the direct and reverse fractional anti-derivatives may not be equal.

Next, we present an illustrative example.

**Example 2.** *Assume a two-dimensional problem where $f(\mathbf{x}) = \|\mathbf{x}\|^{-2}$, $\mathbf{x} \in \mathbb{R}^2$ and $\mathbf{r}(u)$ is a circle with radius $\rho$ and centred at the origin. Suppose that $\alpha \in \mathbb{R}$. In this case,*

$$f(\mathbf{r}(u)) = \rho^{-2},$$

$$\mathbf{r}(u) = \rho\cos(u)\mathbf{e}_1 + \rho\sin(u)\mathbf{e}_2, \quad 0 \le u \le 2\pi,$$

*and:*

$$\|\mathbf{r}'(u)\| = \rho.$$

*Observing $\mathbf{v} = (-\sin u, \cos u)$ and:*

$$f(\mathbf{r}(u) - v\mathbf{v}) = f(\rho\cos u + v\sin u, \rho\sin u - v\cos u) = (\rho^2 + v^2)^{-1},$$

*we derived that:*

$$\begin{aligned}
f^{(-\alpha+1)}_{\mathbf{v}}(\mathbf{r}(u)) &= \frac{1}{\Gamma(\alpha-1)}\int_0^\infty v^{\alpha-2}(\rho^2 + v^2)^{-1}dv \\
&= \frac{1}{\rho^2\Gamma(\alpha-1)}\int_0^\infty v^{\alpha-2}\left(1 + \frac{v^2}{\rho^2}\right)^{-1}dv.
\end{aligned} \tag{48}$$

*Performing the variable change $w = \frac{v^2}{\rho^2}$, we obtained that:*

$$\begin{aligned}
f^{(-\alpha+1)}_{\mathbf{v}}(\mathbf{r}(u)) &= \frac{\rho^{\alpha-3}}{2\Gamma(\alpha-1)}\int_0^\infty w^{\frac{\alpha-3}{2}}(1+w)^{-1}dw \\
&= \frac{\rho^{\alpha-3}}{2\Gamma(\alpha-1)}B\left(\frac{1}{2}\alpha - \frac{1}{2}, -\frac{1}{2}\alpha + \frac{3}{2}\right),
\end{aligned} \tag{49}$$

*where B is the beta function and $1 < \alpha < 3$. Finally:*

$$\begin{aligned}
I^\alpha_{\mathbf{r}}f &= \frac{\pi\rho^{\alpha-3}}{\Gamma(\alpha-1)}B\left(\frac{1}{2}\alpha - \frac{1}{2}, -\frac{1}{2}\alpha + \frac{3}{2}\right) \\
&= \frac{\pi^2\rho^{\alpha-3}}{\Gamma(\alpha-1)\sin\left(\pi\left(\frac{1}{2}\alpha - \frac{1}{2}\right)\right)},
\end{aligned} \tag{50}$$

*with $1 < \alpha < 3$.*

## 5. Conclusions

This paper proposed a definition of the fractional line integral, generalising the concept of the fractional definite integral. It was fully compatible and coherent with the classic line integral. It was based on the concept of the fractional anti-derivative used to generalise the Barrow formula. To define the fractional line integrals needed, the Grünwald–Letnikov and Liouville directional derivatives were introduced and their properties described. The integral was defined first for piecewise linear paths and afterwards for regular curves. It was exemplified and its main properties listed. This integral was intended to be the first

step toward the generalization of Green's, Stokes', and Ostrogradsky–Gauss' theorems. This means that a new step is needed: the fractional surface integral definition.

**Author Contributions:** Conceptualization, M.O.; Formal analysis, G.B. and M.O.; Investigation, G.B. and M.O.; Methodology, G.B. and M.O. Both authors have read and agreed to the published version of the manuscript.

**Funding:** The first author was supported by the Autonomous University of Mexico City (UACM) under the project PI-CCyT-2019-15. The work of the second author was partially funded by Portuguese National Funds through the FCT-Foundation for Science and Technology within the scope of the CTS Research Unit-Center of Technology and Systems/UNINOVA/FCT/NOVA, under the reference UIDB/00066/2020.

**Institutional Review Board Statement:** Not applicable.

**Informed Consent Statement:** Not applicable.

**Data Availability Statement:** Not applicable.

**Conflicts of Interest:** The authors declare no conflict of interest.

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
