# Peer review of "Fractional Line Integral"

_mathematics, doi:10.3390/math9101150_

Round 1

Author Response

see file

Reviewer 2 Report

    In the reviewed paper: Fractional Line Integral the Authors: Gabriel Bengochea and Manuel Ortigueira described in detail the way they conducted their analysis. Manuel Ortigueira created his own school of the fractional calculus which currently is the world leading team (with Machado, J. and Trujillo, J. J.) in this area. In my opinion, he is the main voice of this group.
    The theory presented in the reviewed paper is rather complex. But the Authors present it in a clear way. On the basis of the Grünwald--Letnikov and Riemann--Liouville derivative definitions they defined the Grünwald--Letnikov and Liouville directional derivatives and next for broken line paths they defined the fractional line integrals. Additionally they proved the Theorem 4.
     Additional comments:
    1. Section 2 seems too long. I suggest to pass over the piece 2.4 - it is an academic knowledge.

Author Response

see file

Reviewer 3 Report

In the manuscript, the Authors proposed a definition of fractional line integral and its main properties. In the preceding background, many well-known properties of the classical definite integral have been described.

I would like to ask the Authors to explain the following aspects:

Do the Authors know about the practical applications of the proposed definition of fractional line integral? It is worth adding paragraph about the possible applications of such integral in the modelling of physical processes.

In Eq. (2), the meaning of function $u(/alpha)$ and parameter $m$ should be supplemented.

Eq. (9) contains two sums of $N+1$ terms (each sum). Probably, I think there should be sums of $N$ terms (according to Eq. (7)). Please explain it.

Eq. (32) and the leading line of text – the unknown symbol ‘,’ has been occurred (maybe it's a technical problem).

The text in the paragraph before References is incomprehensible (superfluous). Also the second line after Eq. (53).

“Conclusion” section is missing and should be added.

Author Response

See file

Reviewer 4 Report

The manuscript is interesting and has a solid contribution. I think it is publishable material. The presentation is acceptable as well. However, the paper contains over half of self-citations to the second author's previous works, which is potentially problematic. Please try to check if all these citations are necessary.

Author Response

See file

Round 2

Reviewer 1 Report

see the attached

Author Response

See file

Reviewer 3 Report

The corrections made by the Authors are satisfactory.

But, in the text newly added in the revised version of the manuscript, two aspects (probably editorial errors) have caught my attention - see the PDF file in the attachment.

Author Response

See file

Round 3

Reviewer 1 Report

The reviewer acknowledges the effort made by the authors. Paper is nice and has improved.

Comments on language:

Suggestions on English usage have never intended to be exhaustive but providing some hints to improve/correct minor details. The reviewer’s job should not be pointing out misprints that indeed are annoying for any reader. Proofreading by a native speaker would have been useful.

The authors prefer sub-sections to subsections. Well, that is an option. With all my profound respect, I would dare to suggest looking for the “sub-section” word at

  • https://dictionary.cambridge.org/
  • https://www.collinsdictionary.com/
  • https://www.merriam-webster.com/

Even though, some researchers do indeed write sub-section, anti-derivative and even piece-wise.  

L26: “we require they share” -> “we require them to share”

L30: “Using” -> “By using”

L36: “and main” -> “and its main”

Comments on symbols

The meaning of “u” in P6 line 6 can be guessed to be the Heaviside step function but authors should facilitate the reading of their papers to all readers by explaining the mening of each letter/symbol in each formula..

Remark 3 keeps a statement about lack of interest and concerned readers should like to know why without guessing – not more than some words would avoid that loose end.

Comments on Laplace Transform

The word “Laplace” appears just twice: in L42 as “Laplace Transform”, and in L84 2. as “(bilateral) Laplace” to explain Eq(22) which has no use in the whole paper and provides no context to define the fractional line integral. Btw the sentence after Eq(22), if kept, should also indicate to replace L by F.

The authors give for granted a preconception, bias or background of the reviewer about something not questioned at all as it is the preference of one type of LT over the other that is out of the question. The question is that the paper must be clear for all readers to understand what the authors mean. The fact is that Laplace Transforms (unilateral or two-sided) are not used for the purpose of the paper: to define the fractional integral. The same applies to the recalled Fourier Transform which has no use for the purpose of the paper.

If authors do feel the need of keeping the non-used LT, they should make clear which LT they refer to. From their answer and text in the paper it is unclear if they meant insert or replace just before Eq (21),.

Since two letters do appear in Eq (21), it should be made clear which is constant, and clarify or why Re(s)>0 is added to obtain [15].

Comments on mathematical content

When normalizing a vector, it must be stated or take some caution so that the vector considered is non-null (L88, L98).

Hypothesis of each result must be clear. This does not happen just reading the statement of Theorem 3. If authors understand that this is a follow up of the previous and do not clarify what f represents, then it should not be called Theorem and just follow in plain text what they wish to state.

L98-L102 should be mathematically converted into part of a proof.

Comments on no addressed issues in the previous suggestions

“L18 says that the Green theorem may be generalized but this is not done in the paper. The Conclusion postpones it for the future with announcement of other classical results (Stokes, and Ostrogradski-Gauss). There is no mathematical reason for that postponing once fractal line integral has been developed in this paper and it does not require the concept of surface integral. Indeed Green………. theorems have been presented in fractal form in [11] for boundaries parallel to coordinate axis.”

The authors provide no answer to this request and the paper would improve in interest with that application once Green’s Theorem is recalled in the Introduction as some of the uses and advantages of developing this concept.

The author do keep a high rate of self references (6/15) when there is a lot of work developed in the field that should be referenced in addition to [11], when trying (here announcing) fractional versions of Green’s Theorem such as   

Tatiana Odzijewicz; Agnieszka B. Malinowska; Delfim F. M. Torres. Green’s Theorem for Generalized Fractional Derivatives. Fract. Calc. Appl. Anal., Vol. 16, No 1 (2013), pp. 64–75; DOI: 10.2478/s13540-013-0005-z

Author Response

See file
